materials science

microscopic structure, mesoscopic structure, carbon black, natural rubber, uniaxial tension, fatigue process

**Authors for correspondence:**
Chong Sun
e-mail: sunchongmc@qust.edu.cn
Shipeng Wen
e-mail: wensp@ buct.edu.cn

This article has been edited by the Royal Society of Chemistry, including the commissioning, peer review process and editorial aspects up to the point of acceptance.

# Impact of uniaxial tensile fatigue on the evolution of microscopic and mesoscopic structure of carbon black filled natural rubber

Chong Sun[1,2], Zhongjin Du[2], Selvaraj Nagarajan[2],
Hongying Zhao[3], Shipeng Wen[1], Suhe Zhao[1],
Ping Zhang[2] and Liqun Zhang[1]

[1]Beijing Engineering Research Center of Advanced Elastomers, Beijing University of Chemical Technology, Beijing 100029, People's Republic of China
[2]Key Laboratory of Rubber-Plastics, Ministry of Education/Shandong Provincial Key Laboratory of Rubber-plastics, Qingdao University of Science and Technology, Qingdao 266042, People's Republic of China
[3]Institute of Polymer Materials and Plastics Engineering, Clausthal University, Clausthal-Zellerfeld 38678, Germany

CS, 0000-0001-5476-8929

This investigation addresses the evolution of the microscopic and mesoscopic structures distribution, and micro-defects of carbon black (CB) filled natural rubber (NR) under uniaxial tensile condition during the fatigue process. NR was filled with three different grades of CB in order to understand the impact of the structural degree and specific surface areas of CB and fatigue degree on the Payne effect. It was found that the Payne effect was initially suppressed and then enhanced by increasing the degree of fatigue. The decrease of the storage modulus in the low strain area was attributed to the CB network destruction and the breakdown of the matrix cross-linking network in the early fatigue stage. However, by further increasing the degree of fatigue, the spatial rearrangement of CB aggregates with the orientation of molecular chains between adjacent CB aggregates will results in mechanical reinforcement before the appearance of micro-defects. Moreover, it has been demonstrated that the structural degree of CB has a stronger impact on the mesoscopic structures than the specific surface area of CB during the tensile fatigue process.

# 1. Introduction

Nanofilled rubber composites have been widely used in daily life and industry under dynamic cyclic loading conditions, such as seals, tires and shock absorbers. Owing to the cyclic loading, the internal micromechanical structures of the rubbers will endure microscopic stress, and deteriorate the mechanical properties before the appearance of cracks, which might affect the safety and reliability of the rubber products [1]. Hence, it is of vital significance to study the fatigue properties of elastomers under periodic deformation [2–5].

As a matter of fact, characterization of fatigue has attracted considerable attention in the last three decades [6–12]. Usually, the fatigue failure process involves four main stages: specimen integrity, crack nucleation, crack growth and complete damage. Most of the studies mainly focus on fatigue crack nucleation [13–15] and the influence of different factors on the crack growth process [16–23], such as sample size and branching [17], elastomer formulation [18], etc. Simultaneously, the mechanisms of fatigue crack growth under severe loading are established [24,25]. The fatigue damage is mostly due to cavitation induced by the decohesion between zinc oxides and rubber matrix, resulting from the CB network, as well as oxides decohesion and inherent voids. Tijssens *et al.* [26] proposed a possible mechanism for cross-tie fibril generation in crazes of amorphous polymers. Finite-element calculation demonstrated that cavities grow by local plastic flow, leading to a continuous network of main fibrils interconnected by cross-tie fibrils. In addition, some studies have predicted the fatigue life of rubber products through various analytical approaches [27,28]. Fatigue life prediction is also formulated based on the measured fatigue damage parameters and finite-element analysis [4,29–33]. Flamm *et al.* [34] clarified the influence of very high loads on fatigue life of NR. The fatigue life of elastomers is affected by plenty of factors [35]. There is no doubt that the fundamental investigation is significant for practical application of rubber products. Nevertheless, these studies rarely focus on the process from sample integrity to crack nucleation appearance. In fact, the mechanical performance of the polymer products has significantly deteriorated before the occurrence of crack nucleation, which has a negative impact on the safety of rubber products in real applications.

This work aimed to clarify the nature of structural and property changes from sample integrity to crack nucleation in CB filled NR. The evolution of the micro- and mesoscopic structure of the composites during the fatigue process was investigated by DMA, TEM and NMR. We expected that elucidation of the micro- and mesoscopic structure changes along with the macroscopic properties of rubbers during fatigue would provide theoretical support for the design of new rubbery materials for use under dynamic cyclic loading conditions.

It is worth noting that the term 'fatigue life', which is marked as 'L' in equation (1.1), refers to the fatigue time corresponding to the visible 'cavities' on the surface of the specimen by eyes. (electronic supplementary material, figure S1) The 'fatigue time', which is marked as 't' in equation (1.1), can be used to set the time of experiment. The 'degree of fatigue', which is marked as 'd', is defined as the ratio between the 'fatigue life' and 'fatigue time', which can be described by equation (1.1). That is '0% (without fatigue), 20%, 40%, 60%, 80% and 98%' in §3. We chose 10 samples for each group to test the fatigue life, and then adopted the average as the fatigue life of the sample.

$$d = \frac{t}{L}. \tag{1.1}$$

# 2. Material and methods

## 2.1. Materials

Natural rubber (SVR CV 60) was produced in Vietnam. The CB of N234, N326 and N339 was purchased from CABOT corporation. Sulfur, ZnO, stearic acid and NS (*N*-tert-butylbenzothiazole-2-sulfenamide) were industrial grade products, which were provided by Taicang Guanlian Polymeric Material Co., LTD. The analytical characteristics of the three grades of CB and the formulations of the studied compounds are listed in tables 1 and 2, respectively. The unfilled polymer was used as a reference.

## 2.2. Preparation of CB filled NR composites

The rubber compounding was carried out using an internal mixer; the compounding formulation is given in table 1. Initially, natural rubber was fed into the mixer at a temperature of 60°C with 60 r.p.m. rotor speed. ZnO and stearic acid were added after 50 s, followed by the addition of CB. In the next mixing

**Table 1.** Analytical characteristics fatigue life of CB N234, N326 and N339 [36].

| products | DBPA[a] $(cm^3 100^{-1} g^{-1})$ | CTAB[b] $(m^2 g^{-1})$ | fatigue life of the three composites (h) |
|---|---|---|---|
| N234 | 119 | 125 | 1.61 |
| N326 | 71 | 81 | 2.83 |
| N339 | 121 | 92 | 1.83 |

[a]DBPA absorption reflects the total CB structural degree which comprises both aggregates and agglomerates.
[b]CTAB reflects the external surface area which corresponds to the accessible specific surface area of CB.

**Table 2.** Formulations of the compounds.

| ingredient | Phr[c] |
|---|---|
| natural rubber | 100 |
| ZnO | 5 |
| stearic acid | 2 |
| sulfur | 2.25 |
| NS | 0.7 |
| carbon black | 50 |

[c]Phr (parts per hundred rubber).

step, NS and sulfur were added to the internal mixer at 50°C with a rotor speed of 60 r.p.m., and the mixing process lasted for 5 min below 100°C to avoid scorching. The mixed compound was then masticated on a two-roll mill for about 5 min and stored at 20°C for 24 h. Then rubber compounds were cured at 160°C in a hydraulic press. The vulcanized time was optimized by vulcanization times $t_{c90}$ determined from vulcameter measurements. The samples were cut into dumbbell shapes for mechanical measurement.

## 2.3. Experiments

### 2.3.1. Dynamic mechanical analysis

The dynamic mechanical analysis was performed with an Eplexor 500 N dynamic mechanical analyzer (Gabo, Germany) in tension mode. Payne effect measurements were carried out at a constant frequency of 10 Hz at room temperature with strain amplitudes in the range of 0.001 to 20%. Temperature sweep was measured in a temperature range from −90 to 120°C with a heating rate of 3°C min$^{-1}$ and 300 s of soaking time. The stress relaxation test under strain of 30% was performed at 0°C in tension mode on the same equipment. Temperature-frequency sweep measurement was performed using a DMA242 (NETZSCH, Germany) in dual-cantilever mode at temperatures from -−0 to 100°C. Six frequency points (=0.167, 0.25, 0.5, 1.667, 2.5 and 16.2 Hz) were taken at each temperature to construct master curves according to the time-temperature superposition principle.

### 2.3.2. Morphology characterization

The morphological evolution of the CB aggregates in the specimens before and after the fatigue process was characterized by transmission electron microscopy (TEM). Ultrathin sections of the rubber composite were cut by an ultra-microtome EMFC (Leica, Germany) at a temperature of about −100°C and the sample thickness was about 50–60 nm. TEM pictures were taken by a JSM-2100 transmission electron microscope (JEOL, Japan) with an acceleration voltage of 200 kV.

### 2.3.3. Low-field solid-state NMR

This experiment was carried out on a Bruker mini-spec mq20 solid-state NMR spectrometer at 20 MHz (0.5 T) proton resonance frequency. The 90° pulse has a length of 3.3 μs and the dead time was 11 μs.

**Figure 1.** Strain dependence of the storage modulus for NR filled with 50 phr N234 (*a*), N326(*b*) and N339 (*c*) at different degrees of fatigue; (*d*) ΔE′ as a function of the degree of fatigue. The degree of fatigue is defined as the ratio of fatigue times to the fatigue life of the sample and the fatigue life of three composites is listed in table 1.

Magic-Sandwich Echo (MSE) experiment [37] was performed to obtain a fully recovered free-induction-decay (FID) signal, the initial part of which could be lost due to the receiver dead time in the traditional single pulse experiment.

### 2.3.4. Fatigue condition

Fatigue measurements were performed under uniaxial tensile conditions on a fatigue tester (GOTECH, China) at room temperature with a frequency of 5 Hz. The stretching ratio is 100%.

## 3. Results and discussions

It is worth mentioning here that all the tests are performed immediately when the fatigue process is finished in order to avoid the recovery of the initial modulus when the sample is kept for a period of time.

### 3.1. Evolution of Payne effect during fatigue process

Figure 1*a*–*c* shows the strain dependence of the storage modulus for NR filled with three types of CB at different degrees of fatigue. The preliminary data show a typical phenomenon in nanofilled NR. That is the so-called Payne effect, which can be evaluated by the difference between E′ at low strain and that at high strain denoted by ΔE′ [38]. As shown in figure 1*d*, ΔE′ decreases at first, and then gradually increases as the degree of fatigue increases.

It is clear that the ΔE′ at the early stage (*α*) of fatigue decreases, and then increases gradually (*β*), which is much more pronounced for CB N339 filled composites as shown in figure 1*d*. Note that a systematic

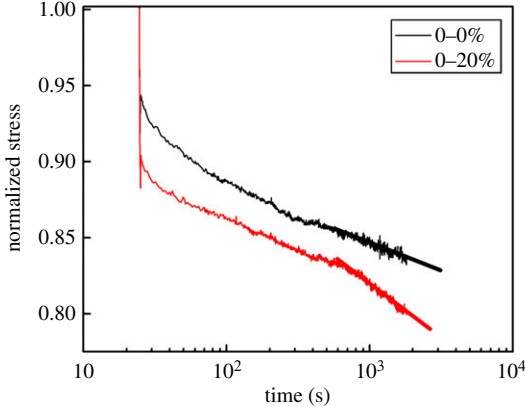

**Figure 2.** The stress relaxation curves for unfilled natural rubber (marked with 0) at fatigue degree of 0 and 20% under strain of 30% at 0°C. The slopes of the two lines at the late stage of relaxation denote the normalized stress relaxation rate $-2.2 \times 10^{-5}$ s$^{-1}$ (black) and $-3.48 \times 10^{-5}$ s$^{-1}$ (red), respectively. The two values arise from linear fitting. According to the tendency of the two lines, the normalized stress relaxation rate of the black one is faster than that of the red one.

reduction of $\Delta E'$ for CB N234 filled compounds before the degree of fatigue at 60% is observed, followed by a slow increase. However, the variation tendency of $\Delta E'$ for the CB N234 filled system is less obvious than that for the CB N339 filled compound. The trend of $\Delta E'$ with an increase of fatigue for the CB N326 filled system is similar to that for CB N234 reinforced composite. The variation amplitude of the $\Delta E'$ for CB N326 filled composite is smaller than for CB N234 filled composite. It is suggested that a possible interpretation of the remarkable destruction of the filler network for the CB N339 filled composite is attributed to its high structural degree. Analytical characteristics of the elastomers were listed in table 1. The matrix around the particles was shielded from CB aggregates, and it increases the effective filler content, indicating a strain-independent contribution to the modulus. [36] Therefore, the storage modulus declines sharply as a result of the disruption of the filler network.

## 3.2. The early stage of fatigue

The decrease of $\Delta E'$ at domain $\alpha$ is principally due to the breakdown of the CB networks. The potential contribution of the destruction of filler–polymer interaction and cross-linking networks may not be excluded from $\Delta E'$.

To further substantiate the role of the cross-linking network at the first fatigue stage, a typical stress relaxation [39,40] for unfilled elastomer is presented in figure 2. The sample is stretched to a 30% strain and maintained at that length about half an hour. A fundamental distinction observed that the normalized stress relaxation rate of unfilled NR at a degree of fatigue of 20% is faster than the sample without fatigue. It was demonstrated that the destruction of chemical bonds during early fatigue progress in the matrix network structure reduces the number of effective chains, maintaining the stress. The matrix cross-linking networks, however, were integral for a specimen before fatigue. Therefore, it can be said that the breakdown of polymer cross-linking networks can also contribute to the drop of $\Delta E'$ at the first stage. The following interpretation of the increased $\Delta E'$ in region $\beta$ is attributed to the spatial rearrangement of filler particles and occurrence of molecular chain orientation between the adjacent aggregates, which is elucidated in detail at a later stage.

Figure 3$a$–$c$ shows the temperature-dependent storage modulus (E′) of composites filled with three grades of CB at various degrees of fatigue.

The storage modulus in the rubbery region decreases with increasing the degree of fatigue. A noticeable drop of the modulus E′ is observed at a degree of fatigue of 20%. This was attributed to a combination of different effects as mentioned above; that is, the destruction of the cross-linking network, macromolecule structures and the breakdown of the filler network, which is in agreement with the results from strain sweep. On the other hand, the storage modulus E′ of the rubbery plateau for CB filled composite gradually decreases with increasing the degree of fatigue in figure 3 inset, because of the much weaker filler–rubber interactions when increasing the degree of fatigue [41].

Figure 3$d$ displays the values for activation energy versus degree of fatigue for three grades of CB filled composites [41]. The activation energy is obtained from the curve by plotting the logarithm of E′ as a function of inverse temperature, which is related to the thermal activation of filler–filler bond. The larger

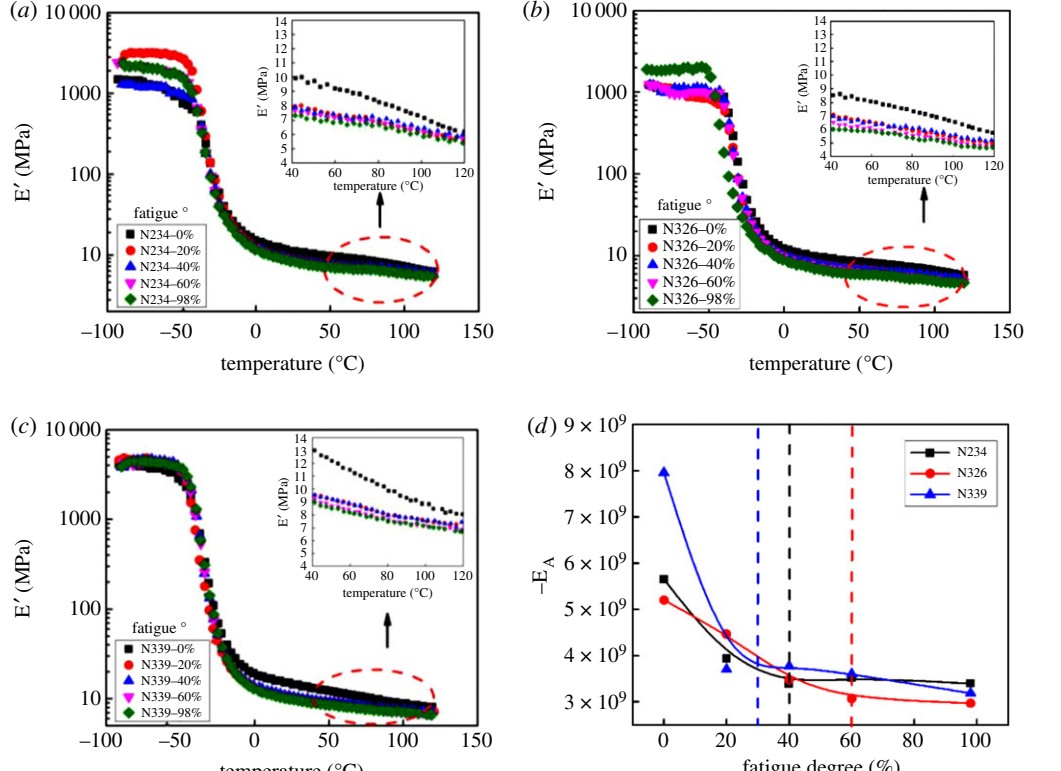

**Figure 3.** The temperature-dependent storage modulus of composites filled with CB N234 (*a*), N326 (*b*) and N339 (*c*) at various degrees of fatigue. Variation of the activation energy as a function of the fatigue degree with three types of CB (*d*).

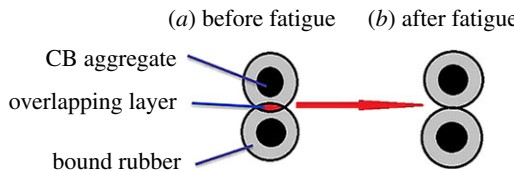

**Figure 4.** Schematic showing the evolution of overlapping layers of bound rubber.

activation energy corresponds to the smaller interparticle distance. It is obvious that the composite filled with CB N339 before fatigue gives the highest activation energy. The smallest interparticle distance between adjacent CB aggregates in CB N339 filled composite, followed by CB N234 and CB N326, respectively. Note that the change of activation energy is remarkable for CB filled composites during the whole fatigue process.

The decrease of the activation energy is ascribed to the fact that the overlapping polymer layers become much thinner [42]. The schematic of overlapping bound rubber layers during the fatigue process is shown in figure 4. The disentanglement of the polymer chains in overlapping layers and the destruction of the chemical bonds leads to a decrease of activation energy, and it reduces the thickness of the overlapping layers. Furthermore, the activation energy of the three composites filled with CB N339, N234 and N326 reached a plateau at a degree of fatigue of 30%, 40% and 60%, respectively, as shown in figure 3*d*. It demonstrates clearly that the destruction of overlapping layers around the adjacent CB aggregates is different.

## 3.3. The late stage of fatigue

Subsequently, the increase of $\Delta E'$ in figure 1*d* $\beta$ region can be explained in more detail. First of all, the morphology evolution of aggregates in the nanocomposites was observed by TEM.

It clearly demonstrated the destruction of the CB agglomerates, which is in good agreement with the decrease of storage modulus in $\alpha$ region of figure 1*d*.

A further finding in figure 5*b* is the aggregate rearrangement in the composite. An interpretation of this behaviour is that the filler–rubber interaction (mainly van der Waals force) is primarily related to the

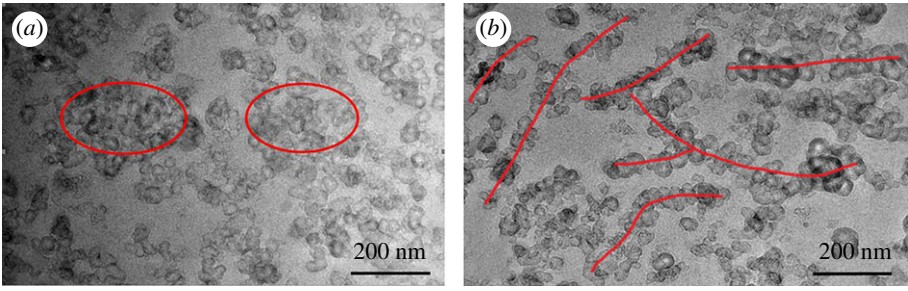

**Figure 5.** TEM images for CB N234 filled natural rubber before (*a*) and after fatigue (*b*).

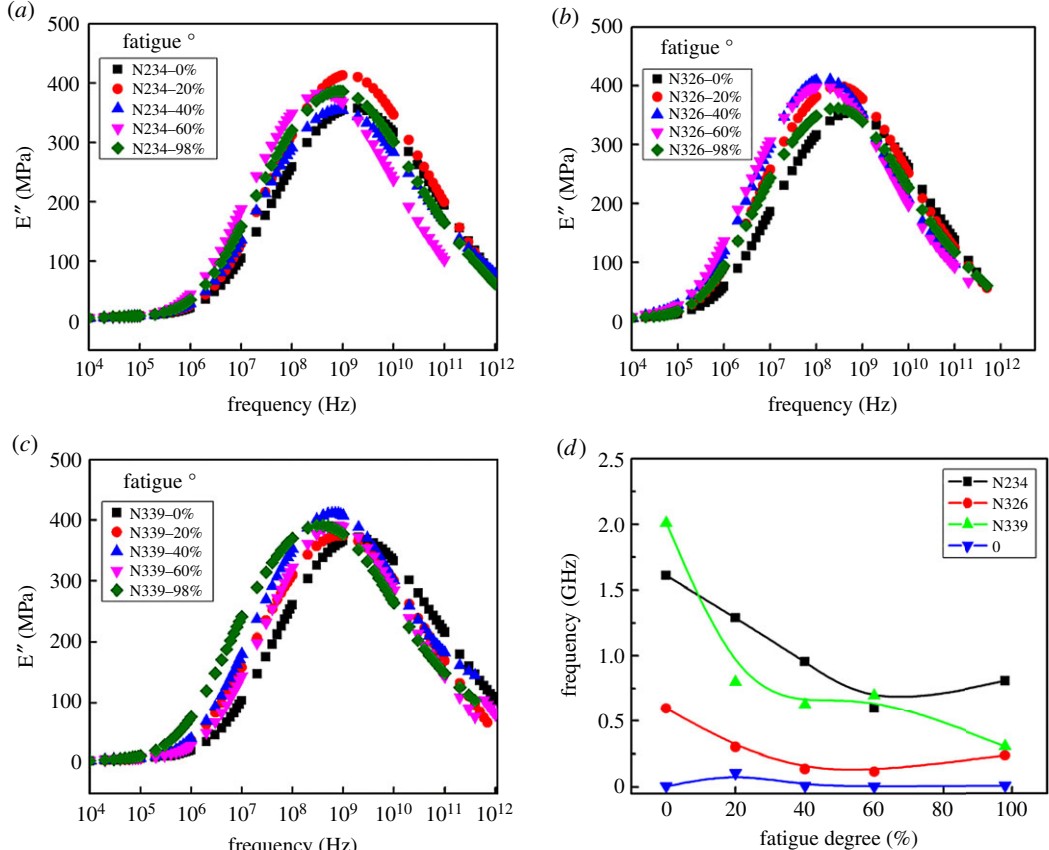

**Figure 6.** Master curves of the loss modulus at 25°C for composites filled with CB N234 (*a*), N326 (*b*) and N339 (*c*) at various degrees of fatigue; (*d*) influence of three different CB and unfilled polymer on glass transition frequency. The glass transition frequency is defined as the peak frequency in the plot of E″.

physical adsorption and desorption [36,43]. The polymer chains of varying lengths, which are adsorbed on the surface of CB aggregates, continuously slip along the CB surfaces during the cyclic loading process, and the filler–rubber interaction was continuously weakened. Meanwhile, the deformation of the polymer chains can also change the filler aggregate network structure in the dynamic fatigue process, leading to the rearrangement [44,45] of CB aggregates in a string. This plays an essential role in reinforcement.

A temperature-frequency sweep measurement was further performed to investigate additional contribution to the increase of ΔE′. The master curves of the loss modulus E″ can be seen in figure 6*a*–*c*, which are constructed by horizontal shifting the E′″-frequency curves according to time-temperature superposition principle [46]. It is interesting to find that the peak frequency position of the loss modulus E″ shifts to the low-frequency direction with the increase of the degree of fatigue.

This means that the polymer segments are more restricted after fatigue. In general, the mobility of the polymer segment should be improved on account of the breakdown of the cross-linking structure and the destruction of filler–polymer interactions. Nevertheless, decreases after fatigue. It is suggested that the orientation of the molecular chains between the neighbouring CB aggregates [46] (i.e. formation of

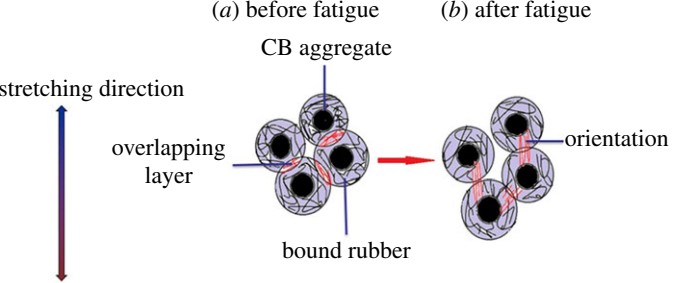

(*a*) before fatigue    (*b*) after fatigue

CB aggregate

stretching direction

overlapping layer

orientation

bound rubber

**Figure 7.** Schematic diagram illustrating the orientation of the molecular chains between adjacent CB aggregates.

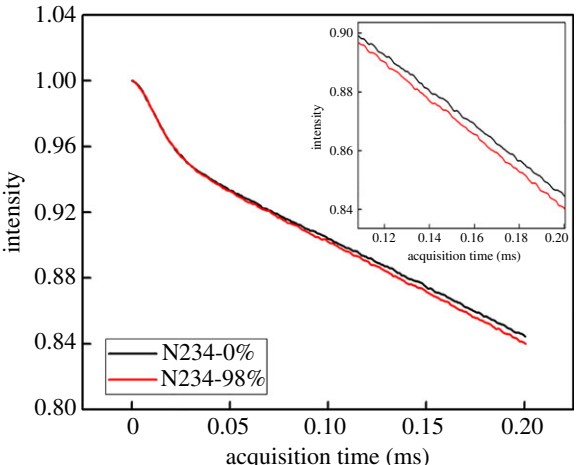

**Figure 8.** MSE-FID signals as a function of acquisition time for composite filled with CB N234 before (N234-0%) and after fatigue (N234-98%).

stretched straight polymer chains between neighbouring CB aggregates [47]) leads to the decrease of glass transition frequency. A schematic diagram of an oriented polymer chain between adjacent aggregates is presented in figure 7.

Therefore, the oriented chains can also increase the $\Delta E'$. Holt *et al.* [48] once concluded that the polymer chains stretching at the interface play a critical role in the suppression of segmental dynamics, and it affects the glass transition temperature. This was consistent with our results that the glass transition frequency gradually decreases with an increase of the degree of fatigue.

Figure 6*d* gives a clear relationship between the glass transition frequency and degree of fatigue for the composites filled with three CB grades and unfilled elastomer. Note that the segmental mobility of the samples with CB N339 and CB N234 are more significantly affected during the fatigue process compared to that of composite filled with CB N326, and the influence of the CB structural degree is greater than the specific surface area. It clearly demonstrated that the high-structure CB N339 corresponds to a developed branched structure. The rubber fills the void space within each aggregate and acts as part of the filler, rather than as part of the deformable matrix [49], which leads to the higher content of occluded rubber. The relatively effective content of the matrix polymer chain is thus decreased, and it leads to the orientation of polymer chains during the fatigue process.

For CB N234 with small particle size, it corresponds to a larger specific surface area. The behaviour of the oriented polymer chain between adjacent aggregates is ascribed to the stronger filler–polymer interaction. Furthermore, the glass transition frequency of these samples before fatigue is distinctly different from figure 6*d*. For the unfilled polymer, the chemical cross-linking density of the matrix is larger than that of the CB filled composites, possibly due to the adsorption of curatives by CB aggregates in a filled system. This leads to a lower glass transition frequency because of the higher cross-linking density. As for CB filled composites, it is suggested that the higher structure and specific surface area have a significant effect on the coating and adsorption of curatives, resulting in the higher glass transition frequency. This is beyond the scope of this work and will not be discussed in detail here.

Based on the findings from a temperature-frequency sweep, further analysis was clarified using Low-Field NMR measurement. The composite reinforced by CB N234 was chosen as the example for

this discussion. Figure 8 exhibited the results of MSE-FID signals. The signal decay rate of the sample after fatigue is faster than that of the sample before fatigue, although the difference between these two samples is not much. However, the relative values are the same within the five repeats. In this case, it is possible to indicate that the polymer chain segment is slightly more restricted after fatigue. It is indirectly manifested that the restriction of chain segment mobility resulted from the stretched polymer chains between adjacent CB aggregates after fatigue.

# 4. Conclusion

In summary, the fatigue behaviour of the NR filled with CB with different morphological parameters such as structural degree and specific surface area was investigated. The Payne effect of three grades of CB versus degree of fatigue was characterized by strain sweep experiments. The E′ at low strain decreases initially and then increases slightly with increasing the degree of fatigue. The whole region of ΔE′-degree of the fatigue plot is divided into region α and β. At region α, the disruption of the filler network plays a dominant role, and the potential contribution of the breakdown of matrix cross-linking network and filler–rubber interaction is also analysed by the stress relaxation and temperature sweep experiments. At region β, the increase of storage modulus with an increase of the degree of fatigue, which is caused by the CB aggregate rearrangement and the orientation of polymer chain, is proved by TEM, DMA and low-field NMR. The spatial rearrangement of CB aggregates is observed by TEM. In addition, the glass transition frequency as a function of the degree of fatigue further demonstrates the reduced segmental mobility of polymer chains after fatigue, which can be ascribed to the orientation of polymer chain between adjacent CB aggregates. The low-field NMR results exhibited enhanced filler–rubber interaction due to the restriction of chain segment mobility, which is in agreement with the above analyses. Furthermore, it can be stated that the effect of structural degree of CB on the evolution of microscopic and mesoscopic structure in CB filled NR is higher than that of the specific surface area of CB.

Data accessibility. Our data are deposited at Dryad Digital Repository: http://dx.doi.org/10.5061/dryad.hf03913) [50].

Authors' contributions. C.S., Z.D. and H.Z. performed the experiments, participated in data analysis and the characterization; C.S. and Z.D. participated in the design of the study and drafted the manuscript; S.N., P.Z. and S.W. coordinated the study and helped draft the manuscript; L.Z. and S.Z. revised the article critically for important intellectual content; all authors gave final approval for publication.

Competing interests. We have no competing interests.

Acknowledgments. The authors gratefully acknowledge the assistance of Dr Rongchun Zhang from South China University of Technology for discussions.

Funding. The work is supported by the National Science Foundation of China (51503114) and (51573007).

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
