## [Reviewer comments · Royal Society Open Science]

Review History

RSOS-181883.R0 (Original submission)

Review form: Reviewer 1 (Yonggang Shangguan)

Is the manuscript scientifically sound in its present form?

Yes

Are the interpretations and conclusions justified by the results?

Yes

Is the language acceptable?

Yes

Is it clear how to access all supporting data?

Yes

Do you have any ethical concerns with this paper?

No

Have you any concerns about statistical analyses in this paper?

No

Recommendation?

Accept with minor revision (please list in comments)

Comments to the Author(s)

The present manuscript investigates the fatigue behaviour of natural rubber filled with three different kinds of carbon black via oscillation amplitude sweep. Reasonable explanation and assumptions corresponds to the structure evolution of CB filled NR during different fatigue stage are proposed through appropriate characterization methods. This work gives new insight into the process from integrity to crack nucleation of filled rubber materials under cyclic loading conditions. The paper can be accepted for publication after minor revisions.

1. "For the unfilled polymer, the cross-linking density is larger than CB filled composites possibly due to the adsorption of curatives by CB aggregates in filled system." (P7) the authors are suggested to supplement the data of swelling equilibrium experiment.

2. According to the analysis of authors, in the early stage of fatigue, disruption of filler network and breakdown of cross-linking network contribute to the decrease of E' , but why would the glass transition frequency also decrease rapidly with low fatigue degree (Fig 6d)?

3. The title of x axis is omitted in Fig 2.

4. The "G" (P5, L3) should be "E".

5. The symbols in Fig 6a should be depicted in forms that can be differentiated more conveniently.

Review form: Reviewer 2 (Richard Windslow)

Is the manuscript scientifically sound in its present form?

Yes

Are the interpretations and conclusions justified by the results?

Yes

Is the language acceptable?

Yes

Is it clear how to access all supporting data?

Yes

Do you have any ethical concerns with this paper?

No

Have you any concerns about statistical analyses in this paper?

No

Recommendation?

Accept with minor revision (please list in comments)

Comments to the Author(s)

I am happy to recommend this paper for publication subject to minor revisions. The paper was well written, made good use of graphics and was generally well researched. There are however a few minor revisions I would like to see prior to publication:

- a) You state that fatigue damage is mostly due to cavitation induced by the decohesion between zinc oxides and rubber matrix.' The paper you cite (Le Cam) actually suggests 4 initiation features; 2 resulting from the Carbon Black network, the third being oxides decohesion and the fourth being inherent voids. I recommend that you adapt the text here to state as such. (Line 52, Page 1)
- b) How effectively can you judge 'visible cavities' particularly if they are 100um? It would be helpful to provide an image of such cavities for the readers to judge. (Line 9, Page 3)
- c) Your definitions for 'degree of fatigue', 'fatigue life' and 'fatigue time' are unclear. Please can you rephrase them make it clearer. It would be useful to have an example to help the reader understand your method. Could the degree of fatigue be shown graphically? (Line 11, Page 3)
- d) Rather than base the fatigue life of the shortest life, would it not have been more appropriate to use the average fatigue life? (Line 13, Page 3)
- e) These are non common filler grades, i.e. N220 or N330 would be more common. Please comment whether there is a specific reason these were chosen. (Line 17, Page 3)
- f) You suggest that the high destruction of the filler network for N339 was due to its high structural degree. Here I suggest that you refer the reader back to your DBPA results for these elastomers. (Line 52, Page 4)
- g) In Figure 2 the x-axis needs labelling. (Line 18, Page 5)
- h) In Figure 2, if you are isolating the viscoelastic aspect of stress relaxation would it not be more appropriate to plot time on a log scale? (Line 18, Page 5)
- i) Units needed for the stress relaxation rates. (Line 21, Page 5)
- j) I realise this may be difficult at this stage, but for the TEM image, Figure 5b, would it be possible to identify the cyclic loading direction on the graph. The figure clearly shows the CB agglomerates aligning but, as is, the alignment direction is unclear. This would help reinforce the point you make at Lines 52-54. (Line 43, Page 6).
- k) You mention that the signal decay rate is faster after sample fatigue; however, the data in Figure 8 is awfully close. Arguably they are within experimental error of each other, which cannot be judge from one test repeat. I suggest you caveat your statement with this point. (Line 36, Page 8)
- l) Correct the power terms for the units such that they are superscript. (Line 5, Page 11)

Dr Richard Windslow
Queen Mary University of London

Decision letter (RSOS-181883.R0)

03-Jan-2019

Dear Professor Sun:

Title: Impact of Uniaxial Tensile Fatigue on the Evolution of Microscopic and Mesoscopic Structure of Carbon Black Filled Natural Rubber
Manuscript ID: RSOS-181883

Thank you for submitting the above manuscript to Royal Society Open Science. On behalf of the Editors and the Royal Society of Chemistry, I am pleased to inform you that your manuscript will be accepted for publication in Royal Society Open Science subject to minor revision in accordance with the referee suggestions. Please find the reviewers' comments at the end of this email.

The reviewers and handling editors have recommended publication, but also suggest some minor revisions to your manuscript. Therefore, I invite you to respond to the comments and revise your manuscript.

Because the schedule for publication is very tight, it is a condition of publication that you submit the revised version of your manuscript before 12-Jan-2019. Please note that the revision deadline will expire at 00.00am on this date. If you do not think you will be able to meet this date please let me know immediately.

Best wishes,
Dr Laura Smith
Publishing Editor, Journals

On behalf of the Subject Editor Professor Anthony Stace and the Associate Editor Professor Claire Carmalt.

RSC Associate Editor:
Comments to the Author:
(There are no comments.)

RSC Subject Editor:
Comments to the Author:
(There are no comments.)

Reviewer comments to Author:
Reviewer: 1

Comments to the Author(s)

The present manuscript investigates the fatigue behaviour of natural rubber filled with three different kinds of carbon black via oscillation amplitude sweep. Reasonable explanation and assumptions corresponds to the structure evolution of CB filled NR during different fatigue stage are proposed through appropriate characterization methods. This work gives new insight into the process from integrity to crack nucleation of filled rubber materials under cyclic loading conditions. The paper can be accepted for publication after minor revisions.

1. "For the unfilled polymer, the cross-linking density is larger than CB filled composites possibly due to the adsorption of curatives by CB aggregates in filled system." (P7) the authors are suggested to supplement the data of swelling equilibrium experiment.

2. According to the analysis of authors, in the early stage of fatigue, disruption of filler network and breakdown of cross-linking network contribute to the decrease of E' , but why would the glass transition frequency also decrease rapidly with low fatigue degree (Fig 6d)?
3. The title of x axis is omitted in Fig 2.
4. The "G" (P5, L3) should be "E".
5. The symbols in Fig 6a should be depicted in forms that can be differentiated more conveniently.

Reviewer: 2

Comments to the Author(s)

I am happy to recommend this paper for publication subject to minor revisions. The paper was well written, made good use of graphics and was generally well researched. There are however a few minor revisions I would like to see prior to publication:

- a) You state that fatigue damage is mostly due to cavitation induced by the decohesion between zinc oxides and rubber matrix.' The paper you cite (Le Cam) actually suggests 4 initiation features; 2 resulting from the Carbon Black network, the third being oxides decohesion and the fourth being inherent voids. I recommend that you adapt the text here to state as such. (Line 52, Page 1)
- b) How effectively can you judge 'visible cavities' particularly if they are 100um? It would be helpful to provide an image of such cavities for the readers to judge. (Line 9, Page 3)
- c) Your definitions for 'degree of fatigue', 'fatigue life' and 'fatigue time' are unclear. Please can you rephrase them make it clearer. It would be useful to have an example to help the reader understand your method. Could the degree of fatigue be shown graphically? (Line 11, Page 3)
- d) Rather than base the fatigue life of the shortest life, would it not have been more appropriate to use the average fatigue life? (Line 13, Page 3)
- e) These are non common filler grades, i.e. N220 or N330 would be more common. Please comment whether there is a specific reason these were chosen. (Line 17, Page 3)
- f) You suggest that the high destruction of the filler network for N339 was due to its high structural degree. Here I suggest that you refer the reader back to your DBPA results for these elastomers. (Line 52, Page 4)
- g) In Figure 2 the x-axis needs labelling. (Line 18, Page 5)
- h) In Figure 2, if you are isolating the viscoelastic aspect of stress relaxation would it not be more appropriate to plot time on a log scale? (Line 18, Page 5)
- i) Units needed for the stress relaxation rates. (Line 21, Page 5)
- j) I realise this may be difficult at this stage, but for the TEM image, Figure 5b, would it be possible to identify the cyclic loading direction on the graph. The figure clearly shows the CB agglomerates aligning but, as is, the alignment direction is unclear. This would help reinforce the point you make at Lines 52-54. (Line 43, Page 6).
- k) You mention that the signal decay rate is faster after sample fatigue; however, the data in Figure 8 is awfully close. Arguably they are within experimental error of each other, which

cannot be judge from one test repeat. I suggest you caveat your statement with this point. (Line 36, Page 8)

l) Correct the power terms for the units such that they are superscript. (Line 5, Page 11)

Dr Richard Windslow
Queen Mary University of London

Author's Response to Decision Letter for (RSOS-181883.R0)

See Appendix A.

Decision letter (RSOS-181883.R1)

14-Jan-2019

Dear Professor Sun:

Title: Impact of Uniaxial Tensile Fatigue on the Evolution of Microscopic and Mesoscopic Structure of Carbon Black Filled Natural Rubber
Manuscript ID: RSOS-181883.R1

It is a pleasure to accept your manuscript in its current form for publication in Royal Society Open Science. The chemistry content of Royal Society Open Science is published in collaboration with the Royal Society of Chemistry.

On behalf of the Subject Editor Professor Anthony Stace and the Associate Editor Professor Claire Carmalt.

RSC Associate Editor
Comments to the Author:
(There are no comments.)

Reviewer(s)' Comments to Author:

Appendix A

Dear Editors and Reviewers:

Thank you for your letter and for the reviewers comments concerning our manuscript entitled: Impact of Uniaxial Tensile Fatigue on the Evolution of Microscopic and Mesoscopic Structure of Carbon Black Filled Natural Rubber (Manuscript ID: RSOS-181883)

The comments are all valuable and very helpful for revising and improving our paper, as well as the important guiding significance to our researches. We have studied comments carefully and have made correction which we hope meet with approval. The revised portions are marked in red in the paper. The main corrections in the paper and the responds to the reviewers' comments are as following:

Responds to the reviewer's comments: Reviewer 1:

1. Response to comment: "For the unfilled polymer, the cross-linking density is larger than that of CB filled composites possibly due to the adsorption of curatives by CB aggregates in filled system." (P7) the authors are suggested to supplement the data of swelling equilibrium experiment.

Response: Thank you for the nice suggestion. The swelling equilibrium experiment is a good method to evaluate the cross-linking density. However, it includes the effect of the chemical cross-linking of the matrix and filler-polymer interaction for filled system. The contribution of the filler-polymer interaction will makes the cross-linking density of the filled system higher. Therefore, we consider to change the sentence as "For the unfilled polymer, the chemical cross-linking density of the matrix is larger than that for the CB filled composites possibly due to the adsorption of curatives by CB aggregates in filled system".

(Reference: Nonentropic Reinforcement in Elastomer Nanocomposites, DOI: 10.1021/acs.macromol.7b00698, page G)

2. Response to comment: According to the analysis of authors, in the early stage of fatigue, disruption of filler network and breakdown of cross-linking network contribute to the decrease of E' , but why would the glass transition frequency also decrease rapidly with low fatigue degree (Fig 6d)?

Response: It is considered that the breakdown of cross-linking network dominates the glass transition frequency, and the degree of disruption of the cross-linking network is strong at low fatigue degree. Therefore, the glass transition frequency decreases rapidly with low fatigue degree.

3. Response to comment: The title of x axis is omitted in Fig 2.

Response: We are very sorry for our negligence. The title of x axis has been added in the modified version.

4. Response to comment: The “G” (P5, L3) should be “E”.

Response: We are very sorry for our incorrect writing, and we have corrected it in the modified version.

5. Response to comment: The symbols in Fig 6a should be depicted in forms that can be differentiated more conveniently.

Response: We have made correction according to the nice comments. The symbols in Fig 6a have been depicted in different forms.

Reviewer: 2

a) Response to comment: You state that fatigue damage is mostly due to cavitation induced by the decohesion between zinc oxides and rubber matrix.' The paper you cite (Le Cam) actually suggests 4 initiation features; 2 resulting from the Carbon Black network, the third being oxides decohesion and the fourth being inherent voids. I recommend that you adapt the text here to state as such. (Line 52, Page 1)

Response: Thank you for your precise correction. We adapt it in the modification version according to the nice recommendation.

b) Response to comment: How effectively can you judge 'visible cavities' particularly if they are 100um? It would be helpful to provide an image of such cavities for the readers to judge. (Line 9, Page 3)

Response: We made the fatigue machine stopped and stretched the sample into 100% strain. In this case, it will be much easy to judge the 'visible cavities' (red circle). The correlative statement will be included in the modified version and the image will be included in the supporting information.

c) Response to comment: Your definitions for 'degree of fatigue', 'fatigue life' and 'fatigue time' are unclear. Please can you rephrase them make it clearer. It would be useful to have an example to help the reader understand your method. Could the degree of fatigue be shown graphically? (Line 11, Page 3)

Response: The "fatigue life", which is marked as "L" in equation 1, refers to the

fatigue time corresponding to the visible "cavities" on the surface of the specimen by eyes. (Fig.1 in supporting information) The "fatigue time", which is marked as "t" in equation 1, can be used to set the time of experiment. The "degree of fatigue", which is marked as "d", is defined as the ratio between the "fatigue time" and "fatigue life", that is, $d=t/L$.

d) Response to comment: Rather than base the fatigue life of the shortest life, would it not have been more appropriate to use the average fatigue life? (Line 13, Page 3)

Response: It is really true as Reviewer suggested that it is more appropriate to use the average fatigue life. In fact, we used the average fatigue life for the experiment. We have revised it in the modified version.

e) Response to comment: These are non common filler grades, i.e. N220 or N330 would be more common. Please comment whether there is a specific reason these were chosen. (Line 17, Page 3)

Response: Yes, the N220 and N330 are very common in rubber industry. However, the N234, N326 and N339 are also used extensively in tire, conveyer belt and other rubber products. Analytical characteristics of carbon black N234, N326 and N339 have been listed in Table 1. The difference between carbon black N234 and carbon black 339 is the CTAB, but with the similar DBPA, i.e. the particle size is obviously different. The difference between carbon black N326 and carbon black 339 is the DBPA, but with the similar CTAB, i.e. the degree of structure is clearly different. In this work, we want to investigate the effect of particle size and degree of structure of carbon black on the fatigue properties. In this case, it is better to keep one parameter ("particle size" or "degree of structure" of carbon black) constantly and the other one ("degree of structure" or "particle size" of carbon black) will be changed to make the result clear.

f) Response to comment: You suggest that the high destruction of the filler net-

work for N339 was due to its high structural degree. Here I suggest that you refer the reader back to your DBPA results for these elastomers. (Line 52, Page 4)

Response: We agree with this comment. The sentence “Analytical characteristics of the elastomers are listed in Table 1” has been inserted in the modified version.

g) Response to comment: In Figure 2 the x-axis needs labelling. (Line 18, Page 5)

Response: We are very sorry for our negligence of the x-axis. It has been modified in the modified version.

h) Response to comment: In Figure 2, if you are isolating the viscoelastic aspect of stress relaxation would it not be more appropriate to plot time on a log scale? (Line 18, Page 5)

Response: Yes, it will be more appropriate to plot time on a log scale and we have corrected it in the modified version.

i) Response to comment: Units needed for the stress relaxation rates. (Line 21, Page 5)

Response: We are very sorry for our negligence of the units and it has been added in the modified version.

j) Response to comment: I realize this may be difficult at this stage, but for the

TEM image, Figure 5b, would it be possible to identify the cyclic loading direction on the graph. The figure clearly shows the CB agglomerates aligning but, as is, the alignment direction is unclear. This would help reinforce the point you make at Lines 52-54. (Line 43, Page 6).

Response: Thank you very much for the good suggestion. In fact, the plane of TEM image is vertical to the cyclic loading direction. We consider that the cross-section of the sample will turn to be narrow and the contractility will induce the CB agglomerates aligning.

k) Response to comment: You mention that the signal decay rate is faster after sample fatigue; however, the data in Figure 8 is awfully close. Arguably they are within experimental error of each other, which cannot be judged from one test repeat. I suggest you caveat your statement with this point. (Line 36, Page 8)

Response: We will modify it in the modified version like this: "The signal decay rate of the sample after fatigue is faster than that of the sample before fatigue, although the difference between these two samples is not much enough. However, the relative values are the same within the 5 repeats. In this case, it is possible to indicate that the polymer chain segment is a bit more restricted after fatigue."

l) Response to comment: Correct the power terms for the units such that they are superscript. (Line 5, Page 11)

Response: We have made correction and corrected the power terms for the units in Fig. 6 in the new version.

We appreciate for Editors/Reviewers' warm work earnestly, and hope that the correction will meet with approval.

Once again, thank you very much for your careful comments and positive suggestions.

Best regards,

Prof. Dr. Chong Sun